# Rapid isometric hamstring strength scores from professional soccer players at pre-season: Positional differences and relationship to peak force

Nicholas Joel Ripley[1]*, Steven Ross[1], Jack Fahey[1], Christopher Bramah[1,2], Paul Jones[1], Paul Comfort[1,3]

1 School of Health and Society, University of Salford, Salford, United Kingdom, 2 Manchester Institute of Health and Performance, Manchester, United Kingdom, 3 School of Medical and Health Sciences, Edith Cowan University, Joondalup, Australia

* n.j.ripley@salford.ac.uk

## Abstract

The assessment of rapid force within a single joint assessment is becoming common practice within elite sport environments. Recent observations have highlighted that rapid force maybe more sensitive to fatigue than peak force within the hamstrings. Yet limited information exists on rapid force production in elite male soccer players. Eighty-nine senior professional men's soccer players (age; $24.2 \pm 5.1$ years, height; $1.83 \pm 0.05$ m, mass; $78.95 \pm 7.53$ kg) participated within the present study they performed two isometric assessments at joint angles of 30° and 90° of hip flexion and knee. They performed three trials with mean and standard deviation determined differences between soccer positions (defender, midfielder and strikers) and differences between limbs were determined via two-way ANOVA with post-hoc analysis. A small overall effect was observed between positions was observed for both the 90:90 ($p = 0.709$, $\eta^2_P = 0.17$), and 30:30 isometric assessments ($p = 0.604$, $\eta^2_P = 0.02$), with trivial pairwise positional differences ($d = 0.00$–$0.03$) observed for relative force at 100 ms and 200 ms. Pearson's correlation coefficients revealed moderate relationships ($r = 0.34$–$0.48$) between rapid force and peak force. Positional differences were only observed when rapid force was made relative to peak force, with midfielders producing greater relative force within 100 ms in comparison to both defenders and forwards, which could be a result of match demands for high-speed running. Using both peak and rapid force would enable accurate reflection of changes through training or fatigue, which could be used to monitor the risk of hamstring injury.

## Introduction

Hamstring strain injuries still present one of the most prevalent injuries within professional soccer [1,2], with injuries typically occurring during the late swing phase in

**Data availability statement:** All relevant data are within the manuscript and its Supporting information files.

**Funding:** The author(s) received no specific funding for this work.

**Competing interests:** The authors have declared that no competing interests exist.

high velocity actions [3]. Previous recommendations to assess eccentric hamstring assessment in order to monitor the risk of hamstring strain injury incidence have received consistent investigation [4,5], with promising results, however, the uptake of the Nordic hamstring exercises remains low [6,7]. As regular monitoring of performance is of high importance due to limited application of pre-season monitoring to in-season injury risk [4], it is important for a monitoring tool to pose minimal risk to athlete with regards to decreases in performance through actual soreness or perceived limitation of performance with negative connotations to performing the Nordic hamstring exercises. This requirement for a monitoring that is quick and can be performed rapidly with large groups and poses minimal risk to performance, has resulted in a rise in isometric hamstring assessments [8–16].

Currently, greater than 50% of practitioners in professional men's soccer have access to force plates [17], enabling a wide array of performance assessments including isometric assessments to be performed. Force plates testing have frequently been used for multi-joint isometric assessments [18], however single joint isometric strength assessments are becoming common place with practice [10–12,14–16]. Variations of isometric hamstring assessments including standing, prone and supine designs have been used within the literature and practice [8–16], with alterations made to set up to assess the hamstrings using different joint positions and ultimately muscle-tendon unit lengths [19,20].

Peak force within the hamstrings has frequently been observed, with consistent reductions identified following fatiguing activity, such as match play, simulated match play or repeated sprint trials [9,14–16]. This has given practitioners support in using force plates for monitoring purposes as they are all sensitive to fatigue in all testing positions [9,14–16]. However, as peak force has a strong relationship to rapid force [10], and diverging characteristics among athletes who may be more or less rapid force dominant could have implication for monitoring [21]. Specifically, these findings indicate that peak force alone might not be providing a true representation of fatigue induced changes in hamstring force generating characteristics, hence further investigation in to measures of rapid force within soccer players is warranted [21]. There is limited research currently observing the effect of a fatiguing activity on rapid force measures. Bettariga et al. [9,22] and Cosio et al. [23] observed reductions in rate of force and torque development within isometric hamstring assessments following a fatiguing activity (sprint interval intervention and competitive soccer match, respectively), with a greater magnitude of change observed within rate of force development (RFD) in comparison to peak force [9,22], highlighting the need to include measures of rapid force in monitoring process.

Currently, no research has been published on the measures of rapid force within isometric hamstring assessments in professional male soccer players, with observations only made in professional female soccer players [10]. With no descriptive data available on the measures of rapid force within professional soccer players, it limits the utility of using these metrics within practice with no comparative data available. Furthermore, as information around normal limb differences and assessing measures of rapid force relative to peak force is currently lacking within this literature clearly

evidences a need for further investigation which may provide more clarity on training needs [21,24]. Hence, the purpose of the present study was to observe measures of rapid and peak force within isometric hamstring assessments of professional soccer players during pre-season to provide practitioners with descriptive data for positional comparisons and limb differences within two isometric hamstring assessments using force plates. It was hypothesized that no positional differences would exist for professional soccer players for rapid or peak force, and when rapid force is made relative to peak force.

## Materials and methods

An observational cross-sectional design was used to provide within session reliability, determine limb differences and identify normative data on professional lower league soccer players within English soccer at the beginning of pre-season. Data collection was across two pre-season period for the 2024−2025 (01/06/2024–4/072024) season and 2025−2026 season (30/06/2025-08/07/2025), with all testing completed between 01/06/2024-08/07/2025. Goalkeepers were excluded from the present study due to their distinct activity profiles and lower hamstring strain injury rates [25,26].

Based on Borg et al. [27], and previously identified intraclass correlation coefficient (ICC) values [8], an expected ICC of >0.80, an alpha error probability $p < 0.05$ and statistical power of 80% a required sample of 30 was identified. The a-priori sample size required to determine the if differences exist between the four positional groups was also performed using G*Power [28], using the average effect size difference in hamstring peak torque between positional groups reported by ($f = 0.26$) [24], an alpha error probability $p < 0.05$ and statistical power of 80% a required sample of 18 per group was calculated.

### Participants

Eighty-nine senior professional men's soccer players from three professional clubs within English League one (age; 24.2±5.1 years, height; 1.83±0.05 m, mass; 78.95±7.53 kg) participated within the present study, identified as tier 3 as suggested by McKay et al. [29] volunteered to participate in the study. The total sample included 38 defenders, 29 midfielders, 22 strikers. Participants were required to have had no hamstring related injuries for ≥6 months prior to taking part. Organizational consent was acquired prior to approaching the participants and all participants provided written informed consent to participate in the study. Ethical approval was granted by the University of Salford institutional ethics committee (2216) in accordance with the declaration of Helsinki 2013. Participants completed the tests prior to their normal training day, following a standardized warm up including squats, lunges, hamstring stretches, leg swings and calf raises.

### Procedures

The 90:90 and 30:30 isometric assessments were measured using a single force plate (Hawkin Dynamics (HD), Westbrook, ME, USA), sampling at 1000 Hz and collected using HD proprietary software, as per previous observations [10,24]. The force plate was placed upon wooden plyometric boxes at an appropriate height for each participant using a goniometer, this was determined by participants lying in a supine position with their knee and hip at 90° of flexion for 90:90 isometric assessment and their knee and hip at 30° of flexion for 90:90 isometric assessment assessed via a goniometer (Figs 1 and 2). Participants heel resting on the force plate with footwear removed. The test was applied unilaterally with the non-testing leg being placed fully extended next to the box and arms placed across the chest.

Three trials for each leg were executed by the participants driving their heel down into the force platform for 3–5 s following three submaximal trials, each trial was collected by the same practitioner who was an accredited strength and conditioning coach and sport scientist with experience in administering force plate assessments. Participants were instructed to remain as still as possible, without initiating a movement for at least a 1-second period to enable calculation of associated force-time data via HD proprietary software. Participants were instructed to '*pull their heel into the force plate as fast and as hard as possible*', while the assessor secured the participants at the hips. Participants were required to repeat

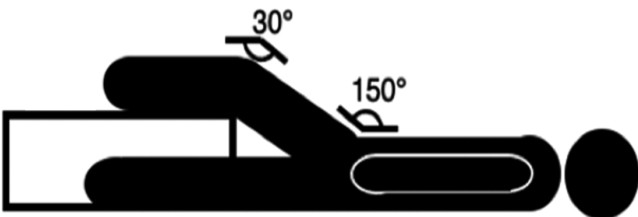

**Fig 1. 30:30 isometric knee flexor test.**

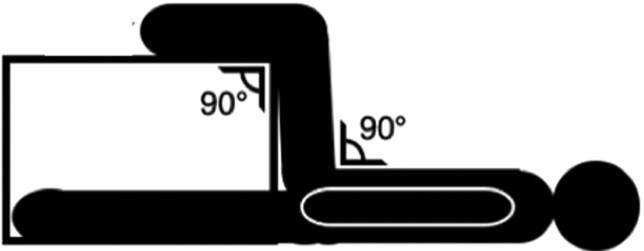

**Fig 2. 90:90 isometric knee flexor test.**

trials if their hips raised off the ground which was determined by visual inspection or if a countermovement was performed, the latter of which was detected through inspection of the force trace following each repetition.

## Data analysis

Force-time data for each trial were automatically analysed using HD proprietary software. Initially force-time data was filtered to 50 Hz using a Butterworth filter automatically applied within the proprietary software for the portable force plates. The HD proprietary software automatically identifies onset of force production using 3 standard deviations (SD) of the stable weighing period this is supported by Guppy et al. [30].

Absolute and relative (ratio scaled to body mass) peak forces and forces at 100 ms and 200 ms were calculated from the force values for each trial, with the average of three trials being taken forward for further analysis. The participants dominant limb (DL) was identified as their kicking limb with the opposite limb identified as non-dominant limb (NDL) [31].

## Statistical analyses

**Within session-reliability.** Data is presented as the mean±SD. Normality was verified using the Shapiro-Wilk's test. A priori alpha level was set at <0.05. Absolute reliability was calculated using coefficient of variance (CV%) based off the sample SD and interpreted based on the upper bound 95% confidence interval (CI) as <5.00%, 5.00–9.99%, 10.00–14.99% and >15% as excellent, good, moderate, and poor, respectively. Relative reliability was assessed using two-way absolute agreement (3,1) ICC [32–34], ICC values were interpreted based on the lower bound 95% CI (ICC; poor <0.49, moderate 0.50–0.74, good 0.75–0.89 and excellent >0.90) as suggested by Koo & Li [34]. The standard error of measurement (SEM) was also calculated as SEM=SD×√1-ICC.

**Differences.** Two-way analyses of variance (ANOVA) with a 3 x 2 mixed design (position x limb) with Bonferroni post-hoc comparisons, bootstrapped to 10,000 samples [35], were conducted to determine whether there were significant and meaningful differences between soccer positions and limbs for relative force at 100 ms, 200 ms and rapid force relative

to peak force (%). Statistical significance was defined as $p \leq 0.05$ for all tests, with resultant $p$ values corrected, using Bonferroni correction, to reduce the risk of a family-wise error. The magnitude of differences within the two-way analysis of variance was also calculated using Partial eta squared ($\eta^2_p$) interpreted as <0.01, 0.01–0.06, 0.07–0.13 and >0.14, as trivial, small, medium and large effect as reported by Cohen [36]. The magnitude of pairwise comparisons were made using Cohen's $d$ effect sizes and interpreted based on the recommendations of Hopkins [37], 0.00–0.19 = trivial and 0.20–0.59 = small, 0.60–1.19 = moderate and >1.20 = large. All statistical analyses were conducted using JASP (Version 0.19.0, computer software).

Pearson's correlations and associated 95% CI bootstrapped to 10,000 samples [35], with coefficient of determination to determine the relationship between absolute and relative peak and time related force values, with paired samples t-test performed to observe the differences between limbs. Correlations were interpreted based on the recommendations of Hopkins [37] 0.00–0.19 = trivial and 0.20–0.29 = small, 0.30–0.49 = moderate, 0.50–0.69 = large, 0.70–0.90 = very large and ≥0.90 = nearly perfect.

## Results

Within session reliability data for peak and rapid force measures are presented in Table 1. All individual data is presented in S1 Table.

Relative peak force values in the 90:90 isometric assessment for defenders, midfielders and forwards were 4.48±0.71 N/Kg, 4.69±0.95 N/Kg and 4.72±0.79 N/Kg, for the for the NDL and 4.55±0.69 N/Kg, 4.63±0.85 N/Kg and 4.79±0.82 N/

Table 1. Within session reliability for peak force and measures of rapid force in the 30:30 and 90:90 isometric hamstring assessment.

| Position and Limb | Metric | CV% (95% CI) | ICC (95% CI) | SEM |
|---|---|---|---|---|
| 90:90 DL | Peak Force (N/Kg) | 5.68 (5.02–6.34) | 0.971 (0.901–0.999) | 0.13 |
| | Force at 100 ms (N/Kg) | 9.02 (7.97–10.07) | 0.709 (0.508–0.967) | 0.42 |
| | Force at 200 ms (N/Kg) | 5.00 (4.42–5.58) | 0.805 (0.646–0.908) | 0.48 |
| | Force at 100 ms Relative to peak force (%) | 10.23 (9.04–11.42) | 0.717 (0.602–0.847) | 8.11 |
| | Force at 200 ms Relative to peak force (%) | 7.89 (6.97–8.80) | 0.792 (0.627–0.897) | 9.33 |
| 90:90 NDL | Peak Force (N/Kg) | 6.11 (5.40–6.82) | 0.975 (0.912–0.998) | 0.13 |
| | Force at 100 ms (N/Kg) | 9.03 (7.98–10.08) | 0.702 (0.523–0.956) | 0.39 |
| | Force at 200 ms (N/Kg) | 5.40 (4.77–6.03) | 0.801 (0.749–0.911) | 0.41 |
| | Force at 100 ms Relative to peak force (%) | 9.11 (8.05–10.17) | 0.753 (0.550–0.836) | 7.29 |
| | Force at 200 ms Relative to peak force (%) | 7.25 (6.40–8.09) | 0.767 (0.622–0.846) | 9.08 |
| 30:30 DL | Peak Force (N/Kg) | 3.39 (3.00–3.79) | 0.980 (0.916–0.998) | 0.13 |
| | Force at 100 ms (N/Kg) | 7.63 (6.75–8.52) | 0.712 (0.537–0.906) | 0.28 |
| | Force at 200 ms (N/Kg) | 5.45 (4.81–6.08) | 0.760 (0.658–0.850) | 0.30 |
| | Force at 100 ms Relative to peak force (%) | 8.73 (7.71–9.74) | 0.789 (0.642–0.901) | 6.83 |
| | Force at 200 ms Relative to peak force (%) | 4.98 (4.40–5.56) | 0.810 (0.670–0.901) | 7.26 |
| 30:30 NDL | Peak Force (N/Kg) | 3.53 (3.12–3.95) | 0.969 (0.895–0.992) | 0.15 |
| | Force at 100 ms (N/Kg) | 7.90 (6.98–8.82) | 0.756 (0.551–0.885) | 0.27 |
| | Force at 200 ms (N/Kg) | 5.30 (4.69–5.92) | 0.786 (0.680–0.981) | 0.31 |
| | Force at 100 ms Relative to peak force (%) | 7.21 (6.37–8.05) | 0.797 (0.640–0.946) | 6.72 |
| | Force at 200 ms Relative to peak force (%) | 6.11 (5.40–6.82) | 0.802 (0.659–0.898) | 7.39 |

CV% = Coefficient of variation percentage; ICC = intra-class correlation coefficient; CI = confidence interval; SEM = Standard error of measurement; DL = dominant limb; NDL = non-dominant limb.

Kg for the DL. Trivial-small differences were observed between positional groups for NDL ($d$ [95%CI] = 0.04–0.30 [−0.96–0.73]) and DL ($d$ [95%CI] = 0.10–0.31 [−0.96–0.89]), with defenders being the weakest positional group.

Relative peak force values in the 30:30 isometric assessment for defenders, midfielders and forwards were 3.86 ± 0.89 N/Kg, 4.10 ± 0.95 N/Kg and 4.01 ± 0.74 N/Kg, for the for the NDL and 4.04 ± 1.02 N/Kg, 4.13 ± 0.85 N/Kg and 4.21 ± 0.99 N/Kg for the DL. Trivial-small differences were observed between positional groups for NDL ($d$ [95%CI] = 0.10–0.28 [−0.93–0.79]) and DL ($d$ [95%CI] = 0.08–0.18 [−0.83–0.77]), with defenders being the weakest positional group.

A small overall effect was observed between positions and all rapid force measures for both the 90:90 ($p$ = 0.709, $\eta^2_P$ = 0.017), and 30:30 isometric assessments ($p$ = 0.604, $\eta^2_P$ = 0.020). Trivial pairwise differences ($d$ = 0.00–0.03) were observed for both relative force at 100 ms and 200 ms (Table 2). Trivial-moderate pairwise differences ($d$ = 0.05–0.71) for

**Table 2. Positional descriptive and pairwise comparisons (Cohen's d effect sizes) for relative values for the 30:30 and 90:90 isometric hamstring assessments and normalized force.**

| | DEF | MID | FOR | Pairwise differences Cohen's d (95% CI) | | |
|---|---|---|---|---|---|---|
| | Mean ± SD | Mean ± SD | Mean ± SD | DEF vs MID | DEF vs FOR | MID vs FOR |
| 30:30 DL Relative force 100 ms (N/kg) | 1.76 ± 0.51 | 1.91 ± 0.50 | 1.68 ± 0.53 | −0.01 (−0.06–0.03) | 0.00 (−0.04–0.05) | −0.02 (−0.07–0.03) |
| 30:30 NDL Relative force 100 ms (N/kg) | 1.62 ± 0.60 | 1.87 ± 0.51 | 1.67 ± 0.50 | −0.02 (−0.07–0.03) | 0.00 (−0.06–0.05) | −0.02 (−0.07–0.04) |
| 30:30 DL Relative force 200 ms (N/kg) | 2.43 ± 0.65 | 2.62 ± 0.58 | 2.44 ± 0.58 | −0.02 (−0.07–0.04) | 0.00 (−0.06–0.06) | −0.02 (−0.08–0.04) |
| 30:30 NDL Relative force 200 ms (N/kg) | 2.33 ± 0.74 | 2.55 ± 0.60 | 2.38 ± 0.60 | −0.02 (−0.08–0.04) | −0.01 (−0.07–0.06) | −0.02 (−0.08–0.05) |
| 90:90 DL relative force 100 ms (N/kg) | 1.95 ± 0.70 | 2.20 ± 0.79 | 1.80 ± 0.69 | −0.02 (−0.08–0.04) | 0.01 (−0.05–0.07) | −0.03 (−0.10–0.04) |
| 90:90 NDL relative force 100 ms (N/kg) | 1.76 ± 0.70 | 1.94 ± 0.67 | 1.80 ± 0.80 | 0.01 (−0.07–0.04) | 0.00 (−0.06–0.06) | −0.01 (−0.08–0.05) |
| 90:90 DL relative force 200 ms (N/kg) | 2.80 ± 1.23 | 2.91 ± 0.99 | 2.58 ± 0.92 | −0.01 (−0.09–0.08) | 0.02 (−0.07–0.11) | −0.03 (−0.12–0.07) |
| 90:90 NDL relative force 200 ms (N/kg) | 2.50 ± 0.85 | 2.80 ± 0.83 | 2.62 ± 1.14 | −0.03 (−0.10–0.05) | −0.01 (−0.09–0.08) | −0.02 (−0.10–0.07) |
| 30:30 DL Relative force 100 ms to Peak force (%) | 46.15 ± 15.77 | 47.32 ± 12.74 | 41.90 ± 15.92 | −0.11 (−1.39–1.18) | 0.38 (−1.02–1.78) | −0.49 (−1.97–0.99) |
| 30:30 NDL Relative force 100 ms to Peak force (%) | 43.15 ± 16.53 | 48.41 ± 17.70 | 41.08 ± 11.29 | −0.47 (−1.76–0.82) | 0.19 (−1.21−1.58) | −0.66 (−2.14−0.82) |
| 30:30 DL Relative force 200 ms to Peak force (%) | 63.40 ± 18.77 | 59.88 ± 15.96 | 64.98 ± 14.30 | −0.14 (−1.58–1.31) | 0.32 (−1.26–1.89) | −0.46 (−2.12–1.21) |
| 30:30 NDL Relative force 200 ms to Peak force (%) | 61.80 ± 18.34 | 65.60 ± 16.65 | 58.55 ± 12.84 | −0.34 (−1.78–1.09) | 0.29 (−1.27–1.85) | −0.63 (−2.29–1.02) |
| 90:90 DL Relative force 100 ms to Peak force (%) | 42.66 ± 15.39 | 47.28 ± 14.05 | 38.54 ± 15.68 | −0.37 (−1.55–0.81) | 0.33 (−0.95–1.61) | −0.71 (−2.07–0.66) |
| 90:90 NDL Relative force 100 ms to Peak force (%) | 39.39 ± 14.44 | 42.03 ± 13.46 | 38.58 ± 16.84 | −0.19 (−1.34–0.96) | 0.09 (−1.16–1.34) | −0.28 (−1.60–1.04) |
| 90:90 DL Relative force 200 ms to Peak force (%) | 61.16 ± 22.53 | 62.97 ± 17.22 | 55.15 ± 20.65 | −0.15 (−1.74–1.45) | 0.49 (−1.26–2.23) | −0.63 (−2.47–1.21) |
| 90:90 NDL Relative force 200 ms to Peak force (%) | 56.64 ± 17.80 | 60.58 ± 15.64 | 55.99 ± 24.13 | −0.32 (−1.80–1.16) | 0.05 (−1.55–1.66) | −0.37 (−2.07–1.33) |

DEF = Defenders, MID = Midfielders, FOR = Forwards, % CI = 95% Confidence intervals, DL = dominant limb, NDL = non-dominant limb, 30:30 = 30° of hip and knee flexion, 90:90 = 90° of hip and knee flexion.

rapid force relative to peak force for both 30:30 and 90:90 isometric hamstring assessments at both 100 ms and 200 ms (Table 2).

Pearson's correlation coefficients revealed moderate relationships between rapid force and peak force (Table 3), stronger correlations were observed for 200 ms for all test positions and limbs. Unsurprisingly, very large to nearly perfect relationships were observed between relative force at 100 ms and 200 ms (Table 3).

## Discussion

Hamstring strain injuries remain highly prevalent within professional soccer [1,2], with many of these injuries occurring within short time frame during the terminal swing phase or early stance phase of sprint running [38,39]. As the hamstrings required to produce extremely high forces rapidly during the proposed mechanisms highlights the need for investigation [40–42]. Therefore, the aim of the present study was to provide descriptive data for rapid isometric force production within two commonly applied hamstring assessments using force plates [9,14–16], determining if there are differences between positional groups and the relationship of rapid force to peak force. In agreement with our hypothesis only trivial differences were observed between positional groups for relative force at 100- and 200 ms (Table 2). However, when rapid force was made relative to peak force, trivial-moderate differences were also identified, which is likely explained by the trivial-small differences present in relative peak force, contrasting the results of an earlier study [24]. Force at 100- and 200 ms was only moderately ($r = 0.35$–$0.48$) correlated to peak force within male professional soccer players, further contrasting previous findings within professional female soccer players [10], which is currently the only available data to compare too using identical methods.

Consistent with previous findings rapid force was reliable, with good-moderate absolute reliability and moderate relative reliability for force at 100- and 200 ms, with force at 200 ms having improved reliability in comparison to force at 100 ms (Table 1). Similar reliability was also observed for rapid force made relative to peak force, with good-moderate absolute reliability and moderate relative reliability. These findings are consistent with previous observations on the reliability of peak and rapid force in isometric hamstring assessments using force plates [8–10,14–16,23,43–45]. Previous authors have observed RFD as a measure of rapid force generating capability, however, previous observations in both the 30:30 and 90:90 isometric assessment have indicated that this metric has poor and unacceptable absolute and relative reliability

**Table 3. Pearson's correlation coefficients and Coefficient of determination between peak and rapid force measures for the 30:30 and 90:90 isometric hamstring assessments for the dominant and non-dominant limbs.**

| Test position | Limb | | r (95% CI) | R² (%) |
|---|---|---|---|---|
| 30:30 | DL | Relative Peak force – Relative force 100 ms | 0.35 (0.17–0.55) | 0.12 (12.25) |
| | | Relative Peak force – Relative force 200 ms | 0.41 (0.27–0.65) | 0.17 (16.81) |
| | | Relative force 100 ms – Relative force 200 ms | 0.90 (0.86–0.93) | 0.81 (81.00) |
| | NDL | Relative Peak force – Relative force 100 ms | 0.36 (0.17–0.53) | 0.13 (12.96) |
| | | Relative Peak force – Relative force 200 ms | 0.45 (0.26–0.64) | 0.20 (20.25) |
| | | Relative force 100 ms – Relative force 200 ms | 0.88 (0.83–0.92) | 0.77 (77.44) |
| 90:90 | DL | Relative Peak force – Relative force 100 ms | 0.45 (0.24–0.62) | 0.20 (20.25) |
| | | Relative Peak force – Relative force 200 ms | 0.48 (0.29–0.64) | 0.23 (23.04) |
| | | Relative force 100 ms – Relative force 200 ms | 0.89 (0.85–0.93) | 0.79 (79.21) |
| | NDL | Relative Peak force – Relative force 100 ms | 0.42 (0.21–0.63) | 0.18 (17.64) |
| | | Relative Peak force – Relative force 200 ms | 0.47 (0.23–0.70) | 0.22 (22.09) |
| | | Relative force 100 ms – Relative force 200 ms | 0.83 (0.75–0.89) | 0.69 (68.89) |

DL = dominant limb, NDL = non-dominant limb, 30:30 = 30° of hip and knee flexion, 90:90 = 90° of hip and knee flexion, r = Pearson's correlation coefficient, R² = Coefficient of determination, CI = confidence interval.

based on the interpretation used within the present study [9,10]. The lower absolute reliability would impact on the utility of RFD as a marker of fatigue, as the increased variability would impact the accuracy of the measurement (i.e., increased measurement error and larger bandwidth for minimal detectable change thresholds). Similarly, the lower relative reliability would mean that making normative benchmarks, as presented for peak force [24], using RFD as a marker of rapid force would be impractical due to the inability for a consistent rank order of participants [10,24]. Therefore, future researchers should look to further investigate the utility of force at set time points (e.g., 50-, 100- and 200 ms) for both fatigue monitoring and benchmarking. These observations would highlight the potential requirement for either rapid or maximal strength training prescription [21,43,46].

Ripley and colleagues [24] have recently shown that when peak force is made relative to body mass (i.e., ratio scaled), there are only trivial differences between positions. This is consistent with the rapid force measures within the present study, with trivial differences for both limbs and positions at 100- and 200 ms (Table 2). However, there were trivial-moderate differences in rapid force relative to peak force, which is likely explained by the differences in peak force. When rapid force was made relative to peak force, midfielders produced the greatest proportion of peak force within 100 ms in comparison to defenders and forwards, with diverging findings observed at 200 ms. These differing findings may be a specific adaptation to positional requirements [47], Fahey et al. [47] identified that within the same leagues as the participants in the present study professional English soccer midfielders were required to run faster than all other positions across 1–10 minute windows within matches [47], with similar observations also noted [48,49]. This finding could highlight the need for midfielders to possess greater early rapid force production capabilities in comparison to other positions, with late rapid force (i.e., 200 ms) and when made relative to peak force not being discriminatory between positions. However, it should be noted that all positions are required to sprint at near maximal intensities during match play the ability to produce force rapidly is an essential quality for all positions.

Previous work from Aagaard et al. [50], Andersen et al. [51] and Oliveira et al. [52] highlighted that monitoring changes in rapid force and observing rapid force as a percentage of peak force enables identification of differential adaptations. A range of potential adaptations through training has been suggested including changes in motor neuron recruitment, motor unit firing frequency, myosin heavy chain isoform composition, sarcoplasmic reticulum calcium kinetics, reduction in type 2x fibres and changes in maximal force production [50–52]. Changes in early and late rapid force production have also been postulated to be explained by changes in fascicle length [53], with increases in fascicle length resulting in reduced early force production capabilities. This finding may have implications upon hamstring strain injury risk reduction practises, where a key aim is to increase bicep femoris fascicle length [5,54,55]. Recently, authors have suggested that observing both rapid and peak force independently and combined as a relative percentage could be used as a method to monitor fatigue [21]. The descriptive data presented within the present study could be used by practitioners to determine the training needs for athletes. For example, if force at 100- or 200 ms or rapid force relative to peak force is substantially below the descriptive values for each positional group this would enable identification of training needs if assessed in a non-fatigued state [21]. However, it is worth noting that the present study was conducted within the first week of pre-season and may not be representative of optimised force production characteristics required by professional soccer players and should be interpreted with caution. Yet the findings were performed in a rested, non-fatigued state meaning the relationship between rapid and peak force may only strengthen with increased strength training as suggested by Aagaard et al. [50]. Moreover, the reliability and ecological validity highlighted within the present study support the use of these assessments and metrics within this population to monitor changes over time which should be the focus of future investigations.

As with previous work in both single joint isometric assessments (including those using the methods used within the present study) and multi-joint isometric force production (e.g., mid-thigh pull) [10,43,46,50,51,56], meaningful relationships between peak and rapid force have been identified. Contrasting the results of the present study, professional female soccer players possessed a large relationship between force at 100 and 200 ms, with rapid force being able to explain 64–79% of the shared variance [10], whereas only moderate relationships were observed for the professional male soccer players

(12–23% shared variance). Comfort et al. [56] observed within the isometric mid-thigh pull, ratio scaling (i.e., relative to body mass) force at 150-, 200- and 250 ms and presenting absolute force at each of these time-points as a percentage of peak force resulted in no meaningful difference between female and male athletes. If the same concept is applied to the isometric hamstring assessments, it could indicate that the professional female soccer players possessed substantially lower relative peak force and rapid force than their male counterparts. However, only absolute net forces were presented previously which limits comparison to the findings of the present study, using the average peak force and body mass reported relative peak force would equate to 3.27 N/Kg supporting the suggestion of strength differences between male and female professional soccer players [10]. The large percentage of shared variance for rapid force to peak force in female soccer players in contrast the male counterparts could be indicative of sex-based difference on rapid force production characteristics. Specifically, female players are producing a large proportion of their available force rapidly, but it is at a lower magnitude of rapid force (i.e., lower relative force at 100 and 200 ms) in comparison to male soccer players. This could be related to genetic or hormonal differences or even related to the social factors around resistance training history and lack of structured resistance training support [57,58]. However, further investigation of these differences is warranted and should attempt to identify the determinants of rapid force production of the hamstrings and attempt to strength match male and female counterparts to further explore the role of peak isometric hamstring force production in rapid force.

We acknowledge that present study is not without its limitations, firstly as the present study was conducted within the first week of pre-season the findings may not be representative of optimised hamstring force production characteristics required by professional soccer players (i.e., peak and rapid) during the season. Future research is warranted to establish if there are differences between phases of the season and acute fatigue monitoring in peak and rapid isometric force within the hamstrings. Moreover, there are a range of methodological questions that need to be answered around the use of force plates in capturing isometric hamstring rapid force measures, specifically with differences in sampling frequency and onset threshold having an effect [8,59,60], with data filtering and cueing currently lacking further investigation.

## Conclusions

Practitioners working within soccer can reliably collect peak and rapid hamstring isometric force data using force plates and by using the descriptive data identified can compare their own athletes to for the identification of individual training needs (i.e., maximal or rapid force production). Positional differences were only observed within rapid force relative to peak force, with midfielders producing greater relative forces within 100 ms in comparison to both defenders and forwards, which could be a result of match demands for high-speed running as identified by Fahey et al. [47]. This could suggest training the hamstring for rapid force production within the first 100 ms is important for midfielders and should be prioritized, however, further investigation is warranted as all players will be required to sprint at near maximal velocities and required to produce high forces rapidly. With only mode moderate relationships between peak and rapid force, practitioners should monitor both metrics in isolation and in combination. Using both peak and rapid force would enable accurate reflection of changes through training or fatigue, which could be used to monitor the risk of hamstring injury.

## Supporting information

**S1 Table. Supporting information with individual data for both the 30:30 and 90:90 isometric hamstring assessments.**
(XLSX)

## Author contributions

**Conceptualization:** Nicholas Joel Ripley, Jack Fahey, Paul Comfort.

**Data curation:** Nicholas Joel Ripley, Jack Fahey.

**Formal analysis:** Nicholas Joel Ripley.

**Investigation:** Nicholas Joel Ripley, Christopher Bramah.

**Methodology:** Nicholas Joel Ripley.

**Project administration:** Nicholas Joel Ripley.

**Writing – original draft:** Nicholas Joel Ripley.

**Writing – review & editing:** Nicholas Joel Ripley, Steven Ross, Jack Fahey, Christopher Bramah, Paul Jones, Paul Comfort.

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
