## [Decision Letter · Decision Letter 0]

17 Mar 2026

Dear Dr. Ripley,

Thank you for submitting your manuscript to PLOS ONE. After careful consideration, we feel that it has merit but does not fully meet PLOS ONE’s publication criteria as it currently stands. Therefore, we invite you to submit a revised version of the manuscript that addresses the points raised during the review process.

We look forward to receiving your revised manuscript.

Kind regards,

Julio Alejandro Henriques Castro da Costa

Academic Editor

PLOS One

Journal Requirements:

https://journals.plos.org/plosone/s/file?id=ba62/PLOSOne_formatting_sample_title_authors_affiliations.pdf....

- DOI: 10.1519/JSC.0000000000005217

In your revision ensure you cite all your sources (including your own works), and quote or rephrase any duplicated text outside the methods section. Further consideration is dependent on these concerns being addressed

Reviewers' comments:

Reviewer's Responses to Questions

**Comments to the Author**

1. Is the manuscript technically sound, and do the data support the conclusions?

Reviewer #1: Partly

2. Has the statistical analysis been performed appropriately and rigorously?

Reviewer #1: No

3. Have the authors made all data underlying the findings in their manuscript fully available?

Reviewer #1: Yes

4. Is the manuscript presented in an intelligible fashion and written in standard English?

Reviewer #1: Yes

Reviewer #1: The article Rapid Isometric Hamstring Strength Scores from Professional Soccer Players at Pre Season: Positional Differences and Relationship to Peak Force presents interesting results, but there are some inconsistences, moreover results, discussion and figures must revised to clarity the manuscript.

Fix the data collection dates, which are currently impossible (listed as 2024–2025). Replace with the accurate preseason-testing window.

Line 96: Methods report data collection “between 01/06/2024–08/07/2025,” which spans beyond the present date and likely includes a year mismatch relative to “first week of pre season.” Please confirm the exact dates and season.

Clarify the ANOVA model, the manuscript alternates between “one way” and “two way” ANOVA. Specify the correct factorial structure and apply it consistently. Moreover, to describe limb factor, the author used mixed test, it is necessary to justify the multiplicity strategy across variables/timepoints. The study examines dominant vs non dominant limbs, yet some analyses use paired t tests (for limb differences) and others suggest a two way ANOVA including limb. Please standardize: if limb is a factor, a two way repeated measures (within subject limb) × position (between subject) model may be most appropriate for the variables of interest.

Consider adding SEM to complement CV% to improve applied interpretation.

Effect size interpretations must be updated. Clarify the exact factorial structure and ensure the reported η²_p (e.g., 0.17 labeled “small”) is consistent with the correct model and accepted magnitude thresholds. Correct effect size labels: η²ₚ=0.17 is labeled “small” but the manuscript’s own thresholds classify it as large.

Force was sampled at 1,000 Hz and low pass filtered with a 4th order, zero lag Butterworth filter at 50 Hz. It is necessary the justification for 50 Hz cutoff, and the consistent onset threshold application (3 SD).

Discussion must be rewrite in order to emphasize that positional differences appear only after normalizing rapid force to peak force, and should therefore be interpreted cautiously.

Tables

Table 2 header says “absolute and relative values” but only shows relative metrics; align title/content.

Figures:

Provide full, descriptive captions for Figures 1 and 2. Must include posture, goniometer use, hip stabilization, and rejection criteria.

.

Reviewer #1: No

---

## [Author Response · Author response to Decision Letter 1]

20 Mar 2026

We have addressed the style concerns.

- DOI: 10.1519/JSC.0000000000005217

In your revision ensure you cite all your sources (including your own works), and quote or rephrase any duplicated text outside the methods section. Further consideration is dependent on these concerns being addressed

We have attempted to address all aspects identified as part of the review regarding the style and overlapping text. If any specific issues relating to the text can be identified and clarified that would be greatly appreciated.

We have now included the caption as required.

We have now included the caption as required.

Dear reviewer thank you for your review, we have made all required changes as requested and believe the manuscript is stronger for the review.

Reviewer #1: The article Rapid Isometric Hamstring Strength Scores from Professional Soccer Players at Pre Season: Positional Differences and Relationship to Peak Force presents interesting results, but there are some inconsistences, moreover results, discussion and figures must revised to clarity the manuscript.

Fix the data collection dates, which are currently impossible (listed as 2024–2025). Replace with the accurate preseason-testing window. Line 96: Methods report data collection “between 01/06/2024–08/07/2025,” which spans beyond the present date and likely includes a year mismatch relative to “first week of pre season.” Please confirm the exact dates and season.

We have added the specific dates to the methods section.

L96-98 – “Data collection was across two pre-season period for the 2024-2025 (01/06/2024-4/072024) season and 2025-2026 season (30/06/2025-08/07/2025), with all testing completed between 01/06/2024-08/07/2025.”

Clarify the ANOVA model, the manuscript alternates between “one way” and “two way” ANOVA. Specify the correct factorial structure and apply it consistently. Moreover, to describe limb factor, the author used mixed test, it is necessary to justify the multiplicity strategy across variables/timepoints. The study examines dominant vs non dominant limbs, yet some analyses use paired t tests (for limb differences) and others suggest a two way ANOVA including limb. Please standardize: if limb is a factor, a two way repeated measures (within subject limb) × position (between subject) model may be most appropriate for the variables of interest.

We have clarified the ANOVA model and factorial structure; the Bonferroni correction was applied to account for the municipality. This is more conservative approach than the Sidak correction, highlighting the strength of our findings.

Consider adding SEM to complement CV% to improve applied interpretation.

We agree with the reviewer, and we have added SEM to compliment the reliability, it is worth noting though that this is based on within session data and should be investigated further.

Effect size interpretations must be updated. Clarify the exact factorial structure and ensure the reported η²_p (e.g., 0.17 labeled “small”) is consistent with the correct model and accepted magnitude thresholds. Correct effect size labels: η²ₚ=0.17 is labeled “small” but the manuscript’s own thresholds classify it as large.

Apologies, this was a typo. This has been corrected in the manuscript. It remains a small effect.

Force was sampled at 1,000 Hz and low pass filtered with a 4th order, zero lag Butterworth filter at 50 Hz. It is necessary the justification for 50 Hz cutoff, and the consistent onset threshold application (3 SD).

Please note this is automatic within the proprietary software, we have clarified this within the text.

L252-153 - “automatically applied within the proprietary software for the portable force plates”

Discussion must be rewrite in order to emphasize that positional differences appear only after normalizing rapid force to peak force, and should therefore be interpreted cautiously.

We have re-ordered the paragraphs within the discussion to follow the statistical approach, reliability, positional differences and then discuss the normalization.

Tables

Table 2 header says “absolute and relative values” but only shows relative metrics; align title/content.

Thank you for noting this we have amended this in the table.

Figures:

Provide full, descriptive captions for Figures 1 and 2. Must include posture, goniometer use, hip stabilization, and rejection criteria.

We have included the individual captions, but we have decided not to include the goniometer use, hip stabilization, and rejection criteria as this is described within the text.

---

## [Decision Letter · Decision Letter 1]

6 Apr 2026

Rapid isometric hamstring strength scores from professional soccer players at pre-season: positional differences and relationship to peak force

PONE-D-26-07511R1

Dear Dr. Ripley,

We’re pleased to inform you that your manuscript has been judged scientifically suitable for publication and will be formally accepted for publication once it meets all outstanding technical requirements.

Kind regards,

Julio Alejandro Henriques Castro da Costa

Academic Editor

PLOS One

Additional Editor Comments (optional):

Reviewers' comments:

Reviewer's Responses to Questions

**Comments to the Author**

Reviewer #1: All comments have been addressed

2. Is the manuscript technically sound, and do the data support the conclusions?

Reviewer #1: Yes

3. Has the statistical analysis been performed appropriately and rigorously?

Reviewer #1: Yes

4. Have the authors made all data underlying the findings in their manuscript fully available?

Reviewer #1: Yes

5. Is the manuscript presented in an intelligible fashion and written in standard English?

Reviewer #1: Yes

Reviewer #1: The authors have adequately addressed my comments raised in a previous round of review. This article is now acceptable for publication.

.

Reviewer #1: No

---

## [Editor Report · Acceptance letter]

PONE-D-26-07511R1

PLOS One

Dear Dr. Ripley,

I'm pleased to inform you that your manuscript has been deemed suitable for publication in PLOS One. Congratulations! Your manuscript is now being handed over to our production team.

Kind regards,

on behalf of

Dr. Julio Alejandro Henriques Castro da Costa

Academic Editor

PLOS One